# Learning Structured Representations by Embedding Class Hierarchy

**Siqi Zeng**
Department of Mathematical Sciences
Carnegie Mellon University
Pittsburgh, PA 15213, USA
`siqiz@andrew.cmu.edu`

**Remi Tachet des Combes**[*]
`remi.tachet@gmail.com`

**Han Zhao**
Department of Computer Science
University of Illinois, Urbana-Champaign
Urbana, IL 61801, USA
`hanzhao@illinois.edu`

## Abstract

Existing models for learning representations in supervised classification problems are permutation invariant with respect to class labels. However, structured knowledge about the classes, such as hierarchical label structures, widely exists in many real-world datasets, e.g., the ImageNet and CIFAR benchmarks. How to learn representations that can preserve such structures among the classes remains an open problem. To approach this problem, given a tree of class hierarchy, we first define a tree metric between any pair of nodes in the tree to be the length of the shortest path connecting them. We then provide a method to learn the hierarchical relationship of class labels by approximately embedding the tree metric in the Euclidean space of features. More concretely, during supervised training, we propose to use the Cophenetic Correlation Coefficient (CPCC) as a regularizer for the cross-entropy loss to correlate the tree metric of classes and the Euclidean distance in the class-conditioned representations. Our proposed regularizer is computationally lightweight and easy to implement. Empirically, we demonstrate that this approach can help to learn more interpretable representations due to the preservation of the tree metric, and leads to better generalization in-distribution as well as under sub-population shifts over multiple datasets.

## 1 Introduction

In supervised learning, the cross-entropy loss is often used for classification tasks. As a common practice in deep learning, in order to train a model for classification, practitioners build a linear layer over the representation to obtain the logit score of each class. A softmax transformation is then applied to convert the logits into a vector belonging to the probability simplex. As a result, we can randomly permute the representations of any classes without affecting the performance of the original classification task. However, in many real-world datasets, as we move towards fine-grained classification, labels are not independent from each other anymore: ImageNet (Deng et al., 2009) inherits label relationship from WordNet (Fellbaum, 1998), that contains both semantic and lexical connections; iNaturalist (Van Horn et al., 2017) borrows the biological taxonomy so that each image contains seven labels that reflect the morphological characteristic of the organism. Many existing works (Deng et al., 2014; Yan et al., 2014; Ristin et al., 2015; Guo et al., 2018; Chen et al., 2019) investigated how to leverage this hierarchical information for various purposes, but how to explicitly project this knowledge onto representations remains unexplored.

In this paper, we focus on the most common label relationship: tree hierarchy. As illustrated in Fig. 1b, given a tree hierarchy of classes, our goal is to learn representations in feature space such

---

[*]Work done while at MSR Montreal.

that the Euclidean distances between different class centers approximate the distances between these classes in the tree. More concretely, we shall first define a *tree metric* to be the length of the shortest path connecting two subset of classes in the tree hierarchy. Based on this tree metric, we then propose a regularizer, the cophenetic correlation coefficient (CPCC) between sequences of tree metric and Euclidean distance of the feature space, to ensure that the class-conditional representations inherit the tree structure of the classes. Different from the original cross-entropy loss with softmax activation, the proposed CPCC regularizer helps to break the symmetry of permutation invariance among the classes, and thus also improves the interpretability of the learned representations.

We show that the proposed CPCC regularizer is computationally lightweight with negligible overhead, and can be applied to a wide range of supervised learning paradigms, including standard flat empirical risk minimization and other hierarchical objectives, including both multitask learning and cirriculum learning. For generalization, over six real-world datasets, we demonstrate that our proposed CPCC regularizer leads to improved generalization performance on some unseen tasks with sub-population shifts when there is only limited amount of labeled data.

## 2 PRELIMINARIES

In this section we first introduce the notations used throughout the paper, formulate our learning problem, and then briefly review the CPCC score to quantify the correlation of two sequences.

**Notations and Setup** We shall use $X$ and $Y$ to denote the input and target random variables, living in spaces $\mathcal{X}$ and $\mathcal{Y}$, respectively. In this work, we mainly focus on the supervised classification setting where for each input data point $x \in \mathcal{X} \subseteq \mathbb{R}^d$, there is a ground-truth label $y \in \mathcal{Y} = [k] := \{1, \ldots, k\}$, where $k$ is the number of output classes. We let $\mu$ be the joint distribution over $(X, Y)$ from where the data is sampled. During the learning process, the learner has access to a dataset $\mathcal{D} = \{(x_j, y_j)\}_{j=1}^n$ of size $n$ sampled from $\mu$.

In the context of representation learning, a learned representation $z = f_\theta(x)$ is obtained by applying a feature encoder $f_\theta : \mathcal{X} \to \mathcal{Z}$ parametrized by $\theta$ to $x$, where $\mathcal{Z} \subseteq \mathbb{R}^p$ denotes the feature space. Upon feature vector $z$, we further apply a linear predictor $g : \mathcal{Z} \to \Delta_k$, where we use $\Delta_k$ to denote the $(k-1)$-dimensional probability simplex. The cross-entropy loss is our objective function. Specifically, let $q_y \in \Delta_k$ be a one-hot vector with the $y$-th component being 1. The cross-entropy loss, $\ell_{\text{CE}}(\cdot, \cdot)$ between the prediction $g \circ f(x)$ and the label $y$ is given by $\ell_{\text{CE}}(g \circ f(x), y) := -\sum_{i \in [k]} q_i \log(g(f(x))_i)$. For $z, z'$, $\|z - z'\|_2$ denotes the Euclidean distance between them.

### 2.1 CLASS HIERARCHY

In classification problems, the target label $Y \in [k]$ is treated as a categorical random variable that can take $k$ different nominal values. However, there is no particular ordering among these $k$ categories, i.e., for different categories $i, j \in [k]$, one can only compare whether $i = j$ or not. Formally, letting $d_H(i, j) = 0$ if $i = j$ and $d_H(i, j) = 1$ otherwise defines a metric $d_H(\cdot, \cdot)$ over $\mathcal{Y}$.

However, in many real-world applications the similarity between different classes is not binary. Consider object classification in ImageNet (Deng et al., 2009) as an example. Intuitively, one would think the distance between the classes corgi and chihuahua to be smaller than that between corgi and panda. One way to characterize this distance between different classes is through a tree of class hierarchy, also known as a dendrogram. An example is shown in Fig. 1a. Formally, let $\mathcal{T} := (V, E, d)$ be a weighted tree, where $V$ is the set of nodes, $E$ the set of weighted edges in $\mathcal{T}$, and $d : V \times V \to \mathbb{R}_+$ specifies the distance between nodes in the tree. Each node $v_S$ in $\mathcal{T}$ is associated with a subset of class labels $S \subseteq [k]$, and can be recursively defined as follows:

1. For each class $i \in [k]$, there is a corresponding leaf node $v_i \in V$. Conversely, each leaf node $v_i \in V$ is identified with a single class label $i \in [k]$.

2. For some $S \subseteq [k]$, if $v_S \in V$ is not a leaf node in $\mathcal{T}$, then its children form a partition of $S$. In other words, if $v_{S_1}, \ldots, v_{S_c}$ are the children of $v_S$, then $\forall i \neq j$, $S_i \cap S_j = \varnothing$ and $\cup_{i \in [c]} S_i = S$.

3. The root node of $\mathcal{T}$ is $v_{[k]}$.

At a colloquial level, the tree $\mathcal{T}$ specifies a hierarchy of class labels that represents the structured knowledge among them. For example, as shown in Fig. 1a of the MNIST dataset, the two children

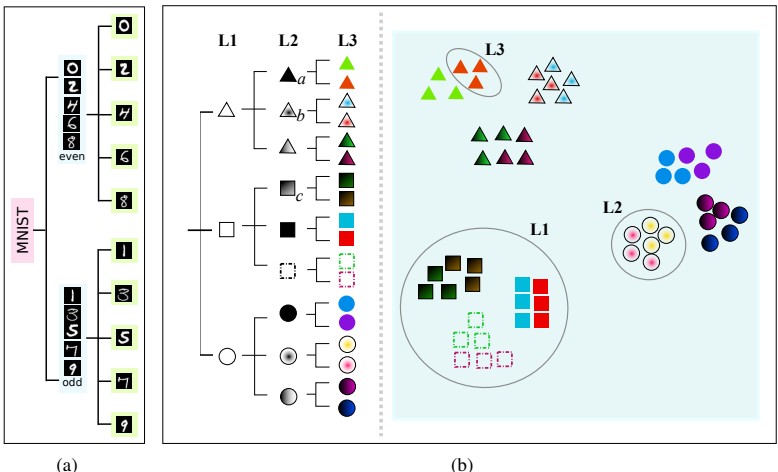

(a)                                    (b)

Figure 1: Fig. 1a: MNIST class hierarchy. The root node contains all the 10 digit classes. The two children nodes of the root node correspond to the *coarse* classes of odd and even digits, respectively. Each leaf node in this class hierarchy corresponds to a *fine* class label (digit). Fig. 1b: An example of a class hierarchy tree $\mathcal{T}$ along with a visualization of the data in the feature space. The CPCC score computes the correlation coefficient of the tree metric from $\mathcal{T}$ in the left panel and the corresponding Euclidean distance obtained from the feature space in the right panel.

of the root node correspond to the odd and even numbers, respectively. Accordingly, the distance between digits 1 and 3 is smaller than that between digits 1 and 2.

## 2.2 COPHENETIC CORRELATION COEFFICIENT (CPCC)

In the context of clustering, Sokal & Rohlf (1962) introduced the cophenetic correlation coefficient (CPCC) to evaluate the correspondence between two dendrograms. The CPCC is the Pearson's correlation coefficient between two sequences of pairwise distances. For a class hierarchy $\mathcal{T}$ and a node $v \in V$, the depth $dt(v)$ of $v$ is the length of the shortest path from $v$ to the root of $\mathcal{T}$. In the original applications of CPCC, the "dendrogrammatic " ground-truth distance $t(v_i, v_j)$ between a pair of nodes $v_i, v_j$ in $\mathcal{T}$ is defined as follows:

$$t(v_i, v_j) := \max\{dt(v_i), dt(v_j)\} - dt(\text{LCA}\,(v_i, v_j)),$$

where LCA $(v_i, v_j)$ is the least common ancestor (LCA) of $v_i$ and $v_j$. In Fig. 1a, the LCA of 1 and 3 is "odd" while the LCA of 1 and 0 is the root node. As an example of ground-truth distance, let us consider the class hierarchy tree $\mathcal{T}$ and clustering of classes given in Fig. 1b. In Fig. 1b, $t(\triangle_a, \triangle_b) = 1$ since they share an LCA $\triangle$ at L1, and $t(\triangle_a, \square_c) = 2$ as they go up 2 levels to meet at the the root node, which is their LCA.

Now consider a dataset $\mathcal{D}$. For a node $v_i \in \mathcal{T}$, since $v_i$ corresponds to a subset of classes, we use $\mathcal{D}_i \subseteq \mathcal{D}$ to denote the subset of data points whose class label belongs to $v_i$. The pairwise distance between $\mathcal{D}_i$ and $\mathcal{D}_j$ is then defined as the Euclidean distance between the center of $\mathcal{D}_i$ and $\mathcal{D}_j$: $\rho(v_i, v_j) := \left\| \frac{1}{n_i} \sum_{x \in \mathcal{D}_i} x - \frac{1}{n_j} \sum_{x' \in \mathcal{D}_j} x' \right\|_2$, where $n_i = |\mathcal{D}_i|$ and $n_j = |\mathcal{D}_j|$ are the number of points in each cluster. Then, the CPCC score $\text{CPCC}(t, \rho)$, between distances $t$ and $\rho$ is defined as:

$$\text{CPCC}(t, \rho) := \frac{\sum_{i<j} (t(v_i, v_j) - \bar{t})(\rho(v_i, v_j) - \bar{\rho})}{(\sum_{i<j} (t(v_i, v_j) - \bar{t})^2)^{1/2} (\sum_{i<j} (\rho(v_i, v_j) - \bar{\rho})^2)^{1/2}}, \tag{1}$$

where $\bar{t} := 2 \sum_{i<j} t(v_i, v_j) \,/\, k(k-1)$, $\bar{\rho} := 2 \sum_{i<j} \rho(v_i, v_j)/k(k-1)$ are the averages of all the pairwise distances for $t$ and $\rho$.

# 3 OUR METHOD

In this section we first define an alternative tree metric used to measure the distance between two nodes in a tree. Then, we proceed to discuss our method that uses the proposed tree metric to learn structured representations when a class hierarchy $\mathcal{T}$ is available during supervised learning.

## 3.1 TREE METRIC

Given a tree $\mathcal{T} = (V, E, d)$ forming a class hierarchy, we formally define the *tree metric* $d_\mathcal{T}$ as:

**Definition 3.1** (Tree Metric). The tree metric $d_\mathcal{T}(v, v')$ for any pair of nodes $v, v' \in V$ is the weighted length of the shortest path in $\mathcal{T}$ connecting $v$ and $v'$.

**Proposition 3.1.** For any undirected weighted graph $\mathcal{G}$, $d_\mathcal{T}(\cdot, \cdot)$ is a well-defined metric over $\mathcal{Y}$, if all edge weights of $\mathcal{G}$ are positive and $\mathcal{G}$ is connected.

Our main motivation to use the tree metric $d_\mathcal{T}$ instead of $t$ as defined in Section 2.2 is two-folds.

First, while each edge in the class hierarchy $\mathcal{T}$ corresponds to a subset relationship, there are other kinds of structured relationships between classes that go beyond the subset relationship. For example, the benchmark dataset MetaShift (Liang & Zou, 2021) also contains a graph to describe the relationship between different subsets of classes. However, in MetaShift the weighted edges do not correspond to the subset relationship as in the case of a class hierarchy, but rather to a similarity/discrepancy measure between them. In this case, the relationship between classes corresponds to an undirected and weighted graph, where the notion of LCA does not apply any more. In fact, even in the case of a tree, the LCA is subject to change depending on which node in the tree is chosen as the root node. However, the tree metric $d_\mathcal{T}$ is invariant to rotations of the tree, and applies to both trees and general graphs as shown in Proposition 3.1.

Second, the definition of $t(\cdot, \cdot)$ does not account for the weights of edges in $\mathcal{T}$. This implies that all the fine-grained classes under a given super-class are the same. However, in many applications, different fine-grained classes may have different proportions or importance, for a given super-class. In these cases, the tree metric is more adapted since it also takes into account the edge weights.

Nevertheless, when $\mathcal{T}$ is an unweighted class hierarchy tree, we have the following relationship between the proposed tree metric $d_\mathcal{T}(\cdot, \cdot)$ and $t(\cdot, \cdot)$:

**Proposition 3.2.** Let $\mathcal{T}$ be an unweighted tree with a fixed root node. Then for any pair of nodes $v_i, v_j \in V$, $d_\mathcal{T}(v_i, v_j) = 2t(v_i, v_j) - |dt(v_i) - dt(v_j)|$.

The proof is deferred to App. A. In particular, Proposition 3.2 states that if the depths of $v_i$ and $v_j$ are the same in $\mathcal{T}$, then our tree metric reduces to twice of $t(v_i, v_j)$. Multiplying a variable by a constant does not affect its correlation with others. Henceforth, in what follows, we propose to use the tree metric $d_\mathcal{T}(\cdot, \cdot)$ in replacement of $t(\cdot, \cdot)$ to compute CPCC.

## 3.2 STRUCTURED REPRESENTATIONS BY EMBEDDING THE TREE METRIC

Now that we have defined our tree metric $d_\mathcal{T}(\cdot, \cdot)$, we are interested in learning representations $f_\theta(\cdot)$, by optimizing the model parameter $\theta$, such that the Euclidean distance between the representations of any pair of class centers $i, j \in [k]$ approximates $d_\mathcal{T}(v_i, v_j)$ in the tree $\mathcal{T}$. Consider a dataset $\mathcal{D} = \{(x_i, y_i)\}_{i=1}^n$ of size $n$ for classification problem over $\Delta_k$. We propose to use the CPCC between $\rho_\mathcal{Z}(\cdot, \cdot)$ and $d_\mathcal{T}(\cdot, \cdot)$ as a regularizer to the cross-entropy loss, resulting in the loss function:

$$\mathcal{L}(\mathcal{D}) = \sum_{(x,y) \in \mathcal{D}} \ell_{\text{CE}}(y, g(f_\theta(x))) - \lambda \cdot \text{CPCC}(d_\mathcal{T}, \rho_\mathcal{Z}), \tag{2}$$

where $\lambda > 0$ is the regularization strength ($\lambda = 1$ in the experiments). The Euclidean distance $\rho_\mathcal{Z}$ is computed in feature space. Concretely, we first apply the encoder $f_\theta$ to $\mathcal{D}$ and obtain a set of points in $\mathcal{Z} \times \mathcal{Y}$: $\mathcal{D}_\mathcal{Z} := \{(f_\theta(x_i), y_i)\}_{i=1}^n$. Then, we partition $\mathcal{D}_\mathcal{Z}$ into $k$ subsets according to the ground-truth labels, and consider the same tree structure $\mathcal{T}$ on $\mathcal{D}_\mathcal{Z}$. Note the negative sign before coefficient $\lambda$ in the above formulation, as we wish to maximize the CPCC score.

In practice, at each iteration during training, since stochastic optimization methods are used, we process a batch of inputs instead of the whole data set $\mathcal{D}$. For each incoming batch, we track the

number of finest classes represented in the batch before any pairwise calculation. When all the inputs in a batch come from the same coarse class, the CPCC score is not well-defined due to the 0 variance of $d_{\mathcal{T}}$. This can happen when the batch size is relatively small. In such cases, we fix the value of the CPCC regularizer to 0 to avoid the numerical division by zero error.

**Time Complexity of the CPCC Regularizer** The computation of our CPCC regularizer is lightweight. For a feature space with $p$ dimensions, for each training iteration, there will be at most $O(p \min(b^2, k^2))$ additional computations, where $b$ is the batch size. Such an overhead is often negligible when compared with the computations needed to train a neural network. In App. B we also provide a brief discussion on the convergence of optimizing the above objective function with SGD.

### 3.3 THE BENEFITS OF STRUCTURED REPRESENTATIONS

In what follows, we describe two potential benefits of learning structured representations with the proposed CPCC regularizer, before providing thorough empirical validation in Section 4.

**Interpretability** As we briefly discussed before, one potential drawback of the representations learned through supervised learning is the lack of interpretability. Recent work (Papyan et al., 2020; Han et al., 2021) have both empirically and theoretically (under certain assumptions) shown that under the cross-entropy loss, when enough training has happened, the learned representations will have reduced variance within each class, and the set of features corresponding to different classes will converge to the so-called simplex equiangular tight frame (ETF). Yet, the vertices of the simplex ETF are symmetric (in the sense of being permutation-invariant), hence the class features do not necessarily reflect the similarities/differences between different classes, even in feature space.

By enforcing the Euclidean distances in feature space between different classes to be close to the tree metric through our CPCC regularization, we attempt to break the symmetry in learning the features. This can potentially lead to more interpretable features, as closer classes (in the sense of the tree metric) are closer to each other in feature space.

**Generalization** Another by-product of structured representations is potentially better generalization both in-distribution when only limited amount of labels is available, or under sub-population shifts (Santurkar et al., 2020). To see this, note that the goal of our CPCC regularizer is consistent with classification accuracy: it essentially pushes data from different classes away proportionally to their distance in the tree. Consequently, for sub-population shifts, if the hierarchy correctly captures coarse-fine relationship, future unseen fine-grained classes from the same coarse category will be further away from those under a different coarse category. This may help generalize to unseen fine-grained classes in zero or few-shot learning.

## 4 EXPERIMENTS

In this section, we apply our proposed method to: (i) study how using CPCC during training affects the representation learnt under various training objectives (Section 4.3), and (ii) see how the learned structured representations can improve generalization (Section 4.4).

### 4.1 DATA

We conduct our experiments on MNIST (Lecun et al., 1998), CIFAR100 (Krizhevsky, 2009), and BREEDS (Santurkar et al., 2020). By using this variety of datasets and hierarchies, we get a comprehensive overview of the usefulness of CPCC as a regularizer. See App. G for the full hierarchies.

**MNIST** contains handwritten digits from 0 to 9. We define odd and even digits to be two *coarse* classes (Fig. 1a). The digits in the leaves are called *fine* classes below. The artificial level based on odd- and even-ness corresponds to concepts that are not visually observable. **CIFAR100** comes with a predefined hierarchy: its coarse level has 20 classes, each containing 5 fine classes (*e.g.*, beaver, dolphin, otter, seal and whale belongs to aquatic mammals). While the hierarchies are semantically meaningful, the coarse level labels are not purely defined by visual similarities. For instance, it is hard to tell the size of an animal from its image (making the coarse classes large omnivores/herbivores and mid-size mammals difficult to distinguish). **BREEDS** is a benchmark

built on ImageNet (Deng et al., 2009). It contains a manually calibrated label hierarchy, based solely on shared visual characteristics. Santurkar et al. (2020) proposed four tasks: *LIVING17*, *ENTITY13*, *ENTITY30*, and *NONLIVING26*. For each, we consider the leaf nodes, which are ImageNet classes, as our fine level classes, and define the coarse levels to be their "superclasses" at different depths. We end up with trees that only contain the root node and two levels of the initial hierarchy, and ignore intermediate relationships for CPCC regularization.

**New Levels** Based on the coarse and fine levels in the hierarchy, we insert a *mid* level between the coarse and fine ones, as well as a *coarser* level between the coarse one and the root node. This results in classes verifying $k_{\text{coarser}} < k_{\text{coarse}} < k_{\text{mid}} < k_{\text{fine}}$. In MNIST, the mid classes are 1,3,5 (odd numbers $\leq 5$), 7,9 (odd numbers $> 5$), 0,2,4 (even numbers $\leq 5$), and 6,8 (even numbers $> 5$). We do not consider a coarser level (it is trivial to train on the root node). In CIFAR100, each coarse level (containing 5 fine classes) is split into arbitrary groups of 2/3 fine classes, creating 40 classes in the mid level. 2 arbitrary coarse classes are merged into 1 coarser label, creating 10 coarser labels. Since BREEDS contains 8 non-root levels in total, and all 4 datasets' coarse levels have a depth $\geq 2$, we use the original hierarchy and let the mid level be one level above the fine classes, and the coarser level be one level above the coarse classes.

**Source & Target Split** We split *BREEDS* into source ($s$) and target ($t$): $s$ and $t$ have the same coarser and coarse labels, but mid and fine classes are different. Following this idea, recall we split MNIST/CIFAR's coarse levels into groups of 2 and 3. We take 60 fine classes as $CIFAR^s$ and the rest as $CIFAR^t$, 6 classes as $MNIST^s$, 4 in $MNIST^t$. Due to this construction, there is only one mid class in each of $CIFAR/MNIST^{s/t}$'s coarse class. On the other hand, *BREEDS*'s coarse classes have many mid children.

## 4.2 BASELINES AND METRICS

As mentioned above, we operate in a fully supervised setting. We denote our neural network as a function $h : \mathcal{X} \rightarrow \Delta_k$. $h \doteq g \circ f_\theta$ can be decomposed into a feature extractor $f_\theta$, and a linear classifier $g$ to which the softmax is applied. Training is performed using the following objectives, on the fine-coarse hierarchy of MNIST, CIFAR, and BREEDS, with and without CPCC as a regularization (see App. C, F for more details). Our baselines include:

- **Flat $\ell_{\text{CE}}$**: training on the fine classes only, without leveraging any hierarchical information.
- **Multi-task Learning**: jointly training a two-headed network to treat fine and coarse as two separate tasks. The loss function is the sum of the cross-entropies on the fine and coarse classification tasks, and we simply set the weight of the two parts to 1.
- **Curriculum Learning**: In the spirit of curriculum learning, we first train on the coarse classes using $\ell_{\text{CE}}$ and use $y_{\text{coarse}}$ instead of $y_{\text{fine}}$. In the second step, we remove the linear classifier and fine tune a new one on the fine level labels with $\ell_{\text{CE}}$ as the loss function.
- **Sum Loss**: We define a hierarchical Sum Loss as $\sum \ell_{\text{CE}}(y_{\text{coarse}}, \mathbf{W}h(x)) + \ell_{\text{CE}}(y_{\text{fine}}, h(x))$, $\mathbf{W}$ is a $k_1$ by $k_2$ matrix representing the relationships in the label tree: if a fine class $i$ belongs to a coarse class $j$, then $\mathbf{W}_{ji}$ is 1, otherwise the entry is set to 0.
- **HXE**: The Hierarchical Cross Entropy (Bertinetto et al., 2020) that replaces the predicted output in $\ell_{\text{CE}}$ with weighted hierarchical class conditional probabilities.
- **Soft**: The soft labels objective (Bertinetto et al., 2020) where labels in $\ell_{\text{CE}}$ are derived from a mapping function to encode class node similarity in $y$.
- **Quad**: The Quadruplet Zhang et al. (2016) multi-task loss which combines $\ell_{\text{CE}}$ with a generalized triplet loss to enforce different margins at different levels of the hierarchy.

**Metrics** To evaluate the representation structure learnt with the various loss functions, as well as the influence of CPCC, we use (i) silhouette scores (Rousseeuw, 1987) to measure the salience of clustering patterns at the coarse level, (ii) CPCC as a metric to measure how the whole representation structure is similar to the fine-coarse hierarchy, (iii) $t$-SNE (Van der Maaten & Hinton, 2008) for visualization of the learnt embeddings in 2D, and (iv) a symmetric distance matrix to evaluate the hierarchical structure. Specifically, we calculate the Euclidean distance between the mean representation vectors for each pair of fine classes. The matrix is organized in a way where fine classes from the same coarse class are grouped together, so that the coarse within-cluster distance is shown around the diagonal while other entries present coarse level between-cluster distances.

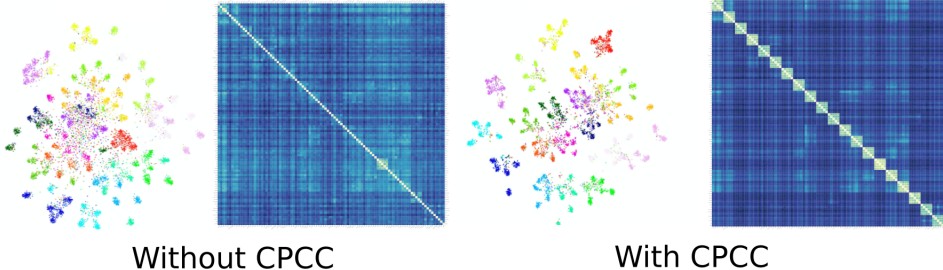

Without CPCC                        With CPCC

Figure 2: The matrices show the distance between fine CIFAR100 classes with and without CPCC for **Flat** (lighter color means smaller distance, same color palette used for both). The light diagonal blocks with CPCC correspond to the coarse classes. We also show $t$-SNE visualization of the representations (colored by coarse labels) learnt using **Flat** with and without CPCC regularization. The pattern is aligned with distance matrix. When CPCC is used, fine classes from the same coarse classes tend to be closer, and coarse classes tend to be further apart.

### 4.3    STRUCTURE OF THE LEARNT REPRESENTATIONS

Fig. 2 show the effects of training with CPCC, which matches our expectations shown in Fig. 1b. In the distance matrix, we can see that the within-coarse cluster distance is much smaller than the between-coarse cluster distance (corresponding to diagonal 5 by 5 blocks). This fact is verified qualitatively in the $t$-SNE plots, where coarse groups tend to be better separated. Similar patterns are observed when CPCC is paired with the other loss functions described above. We want to point out that although other setups have a more structured representation to some extent, these are not as perfect as when paired with CPCC (see App. D for the figures).

In Table 1, we see that the objectives leveraging the hierarchical information tend to increase the CPCC. This is particularly true of the multi-task and soft labels setting. But, directly optimizing the CPCC score still gives the largest gains, both on CPCC and silhouette scores.

### 4.4    GENERALIZATION ON DATASETS WITH A SHARED HIERARCHY

Given a representation $f_\theta$ trained on the fine-coarse hierarchy (Section 4.2), we want to see if structured representations help performance *in-hierarchy*, *i.e.*, on the fine and coarse classes the model was trained on; as well as *out-of-hierarchy*, *i.e.*, on new levels and/or new classes of the hierarchy.

**In-hierarchy** In this setting, we evaluate the models on classes and levels used during training to construct the various objectives and the tree metric (*i.e.*, the fine and coarse classes). Results can be found in the FineAcc and CoarseAcc columns of Table 1. Adding our CPCC regularizer leads to better test accuracy at both levels, across objectives and datasets, with gains sometimes exceeding 1%. According to Goyal et al. (2021), such a performance gain (especially on fine classes) is rarely observed when hierarchical information is leveraged. Overall, our findings suggest that when such information is available, using CPCC as a regularizer is beneficial.

**Out-of-hierarchy** Two questions naturally arise in our hierarchical setting. The first one is how well CPCC structured representations generalize to new levels of the hierarchy. To answer this question, we report in Table 1 the accuracy on the new mid and coarser levels defined in Section 4.1, and not used during training. This accuracy is obtained zero-shot, via a simple marginalization (*e.g.*, the probability of a mid class is the sum of the probabilities of all fine classes that belong to it). There too, adding the CPCC regularization results in performance gains (also see App. Table 4).

The second natural question is if CPCC structured representations can generalize to classes unseen in the training hierarchy. Assume a model has learned that cats and dogs are animals. Does knowing the animal concept help it understand giraffes or horses better? To explore this, we train our models on the source split of the mid and fine classes (Section 4.1), and evaluate their performance on the target split. We can still apply "zero-shot" transfer to coarse and coarser level via marginalization, but fine tuning is necessary to classify the new mid and fine classes: we freeze $f_\theta$ and fine tune a linear classifier $g_{\text{mid}}$ or $g_{\text{fine}}$ on a single image from each new target label (one-shot generalization).

Table 1: Mean % and standard deviation over 5 seeds for various datasets, objectives and metrics, with and without CPCC (overall best in bold, best for a given objective with/without CPCC underlined). *BREEDS*'s results are on the source split. Regularizing with CPCC never hurts performance, and in most cases leads to consistent and sometimes significant improvements on all metrics.

| Dataset | Objective | CPCC | Silhouette | FineAcc | MidAcc | CoarseAcc | CoarserAcc |
|---|---|---|---|---|---|---|---|
| *MNIST* | Flat | 10.80 (1.49) | 13.97 (0.72) | **99.05 (0.23)** | **99.38 (0.04)** | 99.49 (0.08) | N/A |
| | FlatCPCC | **99.96 (0.01)** | **61.33 (0.42)** | 99.28 (0.08) | 99.38 (0.03) | **99.61 (0.03)** | N/A |
| *CIFAR100* | Flat | 24.38 (0.57) | 5.59 (0.02) | 76.82 (0.30) | 80.27 (0.35) | 85.59 (0.35) | 86.85 (0.27) |
| | FlatCPCC | 84.20 (0.39) | 34.40 (0.11) | 77.47 (0.27) | 81.30 (0.14) | 86.95 (0.17) | 88.17 (0.17) |
| | MTL | 39.75 (0.33) | 8.09 (0.08) | 76.56 (0.20) | 80.17 (0.22) | 85.79 (0.20) | 87.11 (0.14) |
| | MTLCPCC | 84.88 (0.58) | 31.58 (0.23) | 76.90 (0.32) | 80.91 (0.29) | 87.11 (0.19) | 88.39 (0.19) |
| | Curr | 23.81 (0.60) | 5.25 (0.11) | 76.84 (0.20) | 80.40 (0.17) | 85.72 (0.16) | 87.02 (0.18) |
| | CurrCPCC | **85.32 (0.51)** | 34.08 (0.23) | **77.48 (0.44)** | **81.42 (0.32)** | **87.15 (0.19)** | **88.44 (0.20)** |
| | SumLoss | 29.85 (0.63) | 4.93 (0.07) | 76.78 (0.20) | 80.47 (0.22) | 85.88 (0.25) | 87.11 (0.26) |
| | SumLossCPCC | 84.78 (0.64) | 31.16 (0.13) | 77.26 (0.12) | 81.17 (0.18) | 86.99 (0.07) | 88.26 (0.02) |
| | HXE | 25.40 (0.68) | 8.31 (0.05) | 76.58 (0.27) | 80.17 (0.24) | 85.67 (0.15) | 87.02 (0.16) |
| | HXECPCC | 85.13 (0.22) | **35.84 (0.18)** | 76.57 (0.33) | 80.63 (0.24) | 86.48 (0.20) | 87.77 (0.20) |
| | Soft | 55.95 (0.67) | 14.48 (0.11) | 76.82 (0.06) | 80.41 (0.07) | 85.84 (0.16) | 87.16 (0.07) |
| | SoftCPCC | 85.23 (0.24) | 35.80 (0.16) | 77.11 (0.16) | 81.02 (0.13) | 86.63 (0.17) | 87.93 (0.14) |
| | Quad | 25.08 (0.26) | 6.75 (0.06) | 76.40 (0.28) | 80.05 (0.27) | 85.30 (0.11) | 86.67 (0.14) |
| | QuadCPCC | 84.65 (0.32) | 34.79 (0.23) | 77.10 (0.16) | 80.92 (0.12) | 86.78 (0.09) | 88.04 (0.09) |
| *LIVING17[s]* | Flat | 36.74 (0.92) | 8.89 (0.08) | 83.66 (0.51) | 89.16 (0.28) | 89.72 (0.26) | 92.54 (0.34) |
| | FlatCPCC | **93.56 (0.46)** | **48.26 (0.51)** | **84.97 (0.68)** | **90.52 (0.54)** | **91.13 (0.61)** | **93.66 (0.36)** |
| *ENTITY13[s]* | Flat | 34.97 (0.43) | 3.12 (0.03) | 82.36 (0.38) | 84.52 (0.33) | 90.43 (0.34) | 93.38 (0.33) |
| | FlatCPCC | **90.98 (0.22)** | **38.68 (0.17)** | **83.37 (0.37)** | **85.53 (0.28)** | **91.50 (0.09)** | **94.22 (0.06)** |
| *ENTITY30[s]* | Flat | 24.29 (0.30) | 3.07 (0.06) | 80.81 (0.35) | 82.28 (0.35) | 86.00 (0.25) | 89.27 (0.27) |
| | FlatCPCC | **73.55 (0.77)** | **29.43 (0.12)** | **82.00 (0.31)** | **83.57 (0.34)** | **87.60 (0.13)** | **90.72 (0.28)** |
| *NONLIVING26[s]* | Flat | 26.95 (0.31) | 5.90 (0.13) | 80.79 (0.36) | 83.49 (0.30) | 80.79 (0.36) | 87.28 (0.25) |
| | FlatCPCC | **82.20 (0.44)** | **34.63 (0.31)** | **82.96 (0.35)** | **85.88 (0.37)** | **87.42 (0.36)** | **89.35 (0.40)** |

Results are shown in Table 2. First, under this subpopulation shift, using CPCC still outperforms the original loss functions on coarse and coarser levels (zero-shot), which is consistent with results in Table 1. Second, in one-shot generalization to new mid levels, CPCC gives an often large advantage. Intuitively, as all fine classes are grouped together within coarse groups, if one data point is randomly selected, then other data points in the same coarse class will readily be assigned the same label. Without this structure in the representation, generalization is more difficult as all fine labels are evenly distributed. The only notable exception is *ENTITY13*, where each coarse label has too many mid level children and grouping by coarse level hurts. Third, CPCC regularization is often harmful to one-shot fine level generalization due to coarse grouping: new fine classes are close together at the coarse level, making them hard to linearly separate. The structure of label tree matters: compared to *LIVING17* and *NONLIVING26*, *ENTITY*'s fine level labels partition coarse labels into much more fine-grained subsets, resulting in the performance difference in *BREEDS*. We do observe that other hierarchical methods have some advantage compared to the flat cross entropy.

## 5 RELATED WORK

There were many works exploiting label hierarchy and we only refer to the most related ones. However, to the best of our knowledge, none of the previous work set learning structured representations as main objective or embedded the tree metric under this context.

**Background of Baseline Methods**  The simplest label hierarchy contains only two level, coarse and fine, which can be treated as two tasks trained jointly or sequentially. The former originates from **Multi-task Learning** (MTL) where part of single network is shared for multiple heads for each task during training (Caruana, 1997; Zhao et al., 2020; Inoue et al., 2020). The latter echoes with **Curriculum Learning** (CL) (Bengio et al., 2009), where pretraining with a easy task will help the convergence and performance on a hard task. We define coarse level classes following Ahn et al. (2021); Peterson et al. (2018); Wang & Cottrell (2015); Stretcu et al. (2021) to emulate human learning behavior. **Sum Loss** directly modifies the loss function by marginalizing fine classes probability for coarse classes. Su & Maji (2021); Hu et al. (2018) defined a similar version to address some partially labeled tasks. Bertinetto et al. (2020) provided a thorough survey of different types of

Table 2: The superscript denotes 1- or 0-shot generalization. All models are trained on the source split $s$ and evaluated on the target split $t$. $s$ and $t$ have different fine/mid classes but the same coarse/coarser classes. CPCC shows an advantage on mid, coarse and coarser, but not fine, levels.

| Dataset | Objective | FineAcc[1] | MidAcc[1] | CoarseAcc[0] | CoarserAcc[0] |
|---|---|---|---|---|---|
| $MNIST^t$ | Flat | **69.93 (10.46)** | 53.25 (4.89) | 51.84 (3.71) | N/A |
| | FlatCPCC | 53.25 (4.89) | **55.11 (8.32)** | **58.75 (2.28)** | N/A |
| $CIFAR100^t$ | Flat | 28.14 (2.50) | 30.66 (3.17) | 42.90 (0.34) | 47.37 (0.46) |
| | FlatCPCC | 25.73 (1.05) | 32.97 (6.55) | 44.58 (0.17) | 48.93 (0.18) |
| | MTL | **29.59 (1.72)** | 30.36 (3.79) | 42.86 (0.31) | 47.34 (0.32) |
| | MTLCPCC | 25.75 (1.61) | 32.47 (5.92) | 44.43 (0.44) | 48.79 (0.43) |
| | Curr | 28.77 (2.73) | 30.82 (4.64) | 43.88 (0.67) | 48.33 (0.65) |
| | CurrCPCC | 25.50 (1.18) | 32.78 (4.47) | 44.65 (0.47) | 48.96 (0.47) |
| | SumLoss | 29.31 (2.62) | 30.62 (3.26) | 43.15 (0.33) | 47.56 (0.37) |
| | SumLossCPCC | 26.39 (1.83) | 32.40 (5.98) | **44.87 (0.37)** | **49.14 (0.39)** |
| | HXE | 28.97 (2.81) | 31.71 (5.51) | 44.28 (0.43) | 48.67 (0.46) |
| | HXECPCC | 25.14 (1.84) | 32.03 (5.68) | 44.38 (0.34) | 48.35 (0.26) |
| | Soft | 29.35 (1.90) | 32.96 (3.31) | 43.99 (0.13) | 48.25 (0.34) |
| | SoftCPCC | 26.10 (1.80) | **34.10 (5.60)** | 44.65 (0.70) | 49.08 (0.68) |
| | Quad | 27.89 (0.24) | 31.28 (4.57) | 42.73 (0.29) | 47.32 (0.54) |
| | QuadCPCC | 24.48 (1.78) | 32.14 (5.57) | 43.68 (0.39) | 48.17 (0.42) |
| $LIVING17^t$ | Flat | 28.52 (3.22) | 32.24 (2.60) | 52.99 (0.79) | 69.68 (0.38) |
| | FlatCPCC | **29.92 (3.40)** | **39.16 (3.30)** | **56.36 (0.69)** | **72.08 (0.43)** |
| $ENTITY13^t$ | Flat | **16.03 (1.60)** | **20.66 (0.43)** | 58.42 (0.33) | 69.76 (0.45) |
| | FlatCPCC | 13.06 (1.64) | 18.89 (1.29) | **61.28 (0.18)** | **71.94 (0.28)** |
| $ENTITY30^t$ | Flat | **21.53 (1.49)** | 21.61 (1.68) | 45.66 (0.20) | 60.07 (0.26) |
| | FlatCPCC | 20.79 (0.78) | **25.29 (1.32)** | **48.83 (0.12)** | **62.89 (0.39)** |
| $NONLIVING26^t$ | Flat | 23.50 (1.74) | 25.37 (2.36) | 39.31 (0.21) | 53.14 (0.14) |
| | FlatCPCC | **24.04 (1.04)** | **27.99 (2.53)** | **42.49 (0.54)** | **56.14 (0.73)** |

hierarchical methods, as well as label embedding methods (encoding hierarchical information into labels, the **Soft** objective), hierarchical losses (**Quad** (Zhang et al., 2016), **HXE**). None of these use a regularization method, with the exception of group overlapping lasso (Zhao et al., 2011). However, it was introduced for logistic regression, making it hard to to be applied to modern neural networks that use the penultimate layer as its representation.

**Learning with Label Hierarchy**   The most common motivation of hierarchical models is to improve the fine-level accuracy. Interestingly, accuracy improvements are often mixed: while most works claimed to gain performance improvement, Wang & Cottrell (2015) stated that this improvement was limited, and Goyal et al. (2021) claimed most hierarchical models lead to worse performance on non-hierarchical accuracy metrics. Additionally, using coarse level labels often appears in a weakly supervised setting, where coarse classes are always available but fine class labels are only accessible for part of data, to reduce annotation cost at a finer level (Taherkhani et al., 2019; Lei et al., 2017; Ristin et al., 2015). Other works built hierarchy from dataset (Murdock et al., 2016; Li et al., 2010; Verma et al., 2012; Han et al., 2018; Zheng et al., 2017). We only name a few since they are very different from our setting where the hierarchy is defined before training.

# 6 CONCLUSION

How to include label relation into representation is an open question. In this paper, in the context of tree label hierarchies, we use the cophenetic correlation coefficient as a regularizer to embed this hierarchical relationship into representations, and outperform other baseline methods. CPCC has multiple advantages, including low time complexity, better interpretability, flexibility on any supervised learning paradigms, and it can be applied to any common label relation graphs. We also demonstrate that it leads to better generalization performance on several downstream tasks. All these benefits show that our method provides an interesting solution to this important problem.

ACKNOWLEDGMENTS

Han Zhao would like to thank the support from a Facebook research award and Amazon AWS Cloud Credits.

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
