# OpenReview forum: "Learning Structured Representations by Embedding Class Hierarchy"
_ICLR.cc/2023/Conference — ICLR 2023 poster_

### Official Review · Reviewer_siuS · 2022-10-21

**Confidence:** 3
**Correctness:** 4
**Technical Novelty And Significance:** 4
**Empirical Novelty And Significance:** 4
**Recommendation:** 8

**Clarity, Quality, Novelty And Reproducibility:**

Clarity:
- This paper is clear and well structured.
Quality:
- The explanation of the main method, the figures and the appendix are proofs of a high quality report.
Novelty:
- A single contribution being the regularization term for structured image classification.
Reproducibility:
- This paper looks possible to replicate.

**Details Of Ethics Concerns:**

No Ethic concern

**Strength And Weaknesses:**

Strength:
- This paper is very well-writtent and well motivated.
- The authors discuss the limitation of their method and provide arguments on results that do not outperform the state-of-the-art.
- The figures are very informative and provide a good insight on the ideas that motivated this work.
- The utilization od CPCC as a regularizer seems novel.
- Strong experimentation showing the strength of the method on a wide variety of datasets.

Weakness:
- There is only one contribution, while it is well detailed, it could still qualify as an incremental improvement to existing work.
- The lack of interpretability is a problem, even if it is discussed by the authors many times in the paper. For example in the MNIST dataset, the authors mentionned that the classes are sorted to the nodes as evens and odds parent classes, and that there is no visual similarity between the leaf node from the same parent. The question that arises is : Is it critical that the tree structure follow some logic ? It would have been interesting to see results on a random tree hierarchy and see if the results are the same. If they are not, then, simply breaking the class permutation invariance seems advantageous. Clearly for other datasets particularly with animals classes, visual similarity seems to be a factor in the tree hierarchy so why is it important in some cases and not others?
- There is also a concern regarding the tree structure and current capacity of deep neural networks. The tree metric is said to be weighted with respect to edge weights in section 3.1 but the difference in magnitude between this tree distance function and the euclidean distance between deep features seems to be on completely different scales. How is this addressed in the regularization function? Is there some additionnal normalization term?

**Summary Of The Paper:**

This paper is about structured representation learning for image classification at different semantic level of granularity. The author proposed to leverage class hierarchy and relationships by introducing a regularization function using the CPCC in addition to the cross-entropy loss. They introduced a measurement of distance between classes structured in a tree and enforce the network to follow the rules of this distance metric in the deep space. They perform on 6 image classification datasets with different structured representation learning paradigm such as multi-task learning, curriculum learning and others.

**Summary Of The Review:**

The related work and citation seems to be accurate and recent. The figures are informative. I easily tend to accept this paper because of the quality of the explanation and the thoughrough experiment section. What would make this paper a candidate for being highlighted would have been an additional contribution.

---

> ### Author Response · Authors · 2022-11-17
> **Response to Reviewer siuS**
>
> *We thank the reviewer for acknowledging our contribution. We next provide clarification to answer your questions:*
>
> > The lack of interpretability is a problem ...  Is it critical that the tree structure follow some logic? It would have been interesting to see results on a random tree hierarchy and see if the results are the same. If they are not, then, simply breaking the class permutation invariance seems advantageous. Clearly for other datasets particularly with animals classes, visual similarity seems to be a factor in the tree hierarchy so why is it important in some cases and not others?
>
> *On MNIST we studied a new hierarchy, by setting 0-4 | 5-9 as the coarse levels instead of splitting by oddness of digit. The fine and coarse accuracy are 99.31 (0.08) and 99.53 (0.04) which still outperforms cross entropy baselines. Here the advantage is more likely to be caused by using extra coarse level of information. Also, increased coarse margin rules out some incorrect labels at the coarse level. If your test set has the same hierarchy as the train set, this information by breaking the class permutation is helpful.*
>
> *Visual similarity is always important for unseen fine class transfer tasks. Intuitively, if the coarse level is well-defined (visually meaningful hierarchies, no vehicles_1/vehicles_2), or the coarse level is close to the fine level (apple and banana are fruit vs. are items; LIVING/NONLIVING vs. ENTITY in our experiments), it’s easier to learn the correct feature that does transfer to classification of unseen fine labels.*
>
> > There is also a concern regarding the tree structure and current capacity of deep neural networks. The tree metric is said to be weighted with respect to edge weights in section 3.1 but the difference in magnitude between this tree distance function and the euclidean distance between deep features seems to be on completely different scales. How is this addressed in the regularization function? Is there some additional normalization term?
>
> *The CPCC is by definition a correlation coefficient, so both distances are automatically normalized and the final outcome will always be in [-1,1]. In our experiments, we set the regularization factor $\lambda$ to 1  (and did not sweep on it). There are no other hyperparameters.*

---

> > ### Comment · Reviewer_siuS · 2022-11-23
> > **Response to authors**
> >
> > I thank the authors for their repsonse and the additionnal experiments, explanations and clarifications. I am keeping the current rating.

---

> > ### Comment · Reviewer_siuS · 2022-11-23
> > **Response to authors**
> >
> > I thank the authors for their repsonse and the additionnal experiments, explanations and clarifications. I am keeping the current rating.

---

### Official Review · Reviewer_NCZE · 2022-10-24

**Confidence:** 4
**Correctness:** 3
**Technical Novelty And Significance:** 2
**Empirical Novelty And Significance:** 2
**Recommendation:** 6

**Clarity, Quality, Novelty And Reproducibility:**

* The paper is well written.

* Deep representation learning with CPCC score as a regularization might be new, but the lack of comparison, both in related work and empirically, makes it difficult to evaluate its novelty.

* The paper provides sufficient details for reproducibility.

**Strength And Weaknesses:**

* Strength
  - The paper is clearly written.
  - The formulations are clean and well motivated.
  - Experimental results show that the method works as expected. Results on one-shot generalization particularly highlights the effectiveness of learning with CPCC score to embed label hierarchy.

* Weakness
  - While I do not closely follow this topic of learning with label hierarchy, it seems there are multiple previous works on deep representation learning with label hierarchy. For example, [[Zhang et al., 2016](https://arxiv.org/pdf/1512.02895.pdf)], [[Bertinetto et al., 2020](https://openaccess.thecvf.com/content_CVPR_2020/papers/Bertinetto_Making_Better_Mistakes_Leveraging_Class_Hierarchies_With_Deep_Networks_CVPR_2020_paper.pdf)]. In the current writeup, the related work on learning with label hierarchy is too short and narrow. Also the baseline considered seems a bit too weak and naive.
  - Experiments on one-shot generalization is not particularly interesting nor significant. For example, in experiment authors introduced "coarser" or "mid" level, which are both coarser level of "coarse" and "fine" class hierarchy. Even though one can train a new classifier using one data point from with "coarser" or "mid" level labels, aren't we supposed to do it in a "zero-shot" way by simply leveraging "coarse" and "fine" grained classifiers (i.e., for mid-level classifier, one can simply define a rule on fine-grained classifier)? Generally, I do not think this is an "one-shot" problem since 1) "coarser" and "mid" level labels can be derived from the "coarse" and "fine" level labels (e.g., "mid" level labels of odd & <= 5 is a union of 1, 3, 5 at fine-grained level labels), 2) many "coarse" and "fine" level labels are given during the model training already.
  - In this regard, more meaningful experiments would be one-shot or few-shot generalization to unseen classes, i.e., can we learn a fine-grained representation of class "0" by leveraging knowledge of its superclass ("even") and neighbors ("2, 4, 6, 8")?

**Summary Of The Paper:**

The paper presents CPCC loss for learning deep representation with label hierarchy. It is based on the CPCC score, while re-defining the distance metric, as a shortest path length between two nodes. Overall proposed method sounds reasonable and looks simple. In experiments, the proposed method is evaluated on MNIST (with odd/even as coarse class), CIFAR-100 (with pre-defined coarse and fine-grained label hierarchy) and BREEDS, with a few baseline models including multi-task of coarse and fine-grained loss, curriculum learning, etc.

In experiments, authors evaluate a few metrics -- shillouette score, CPCC score, t-sne, etc -- to evaluate learned structure. W.r.t. defined metrics, the proposed method showed improved scores over baseline methods. In addition, the paper presents one-shot generalization experiments and demonstrate effectiveness on knowledge transfer.

**Summary Of The Review:**

The paper is very well written and clear. The idea of using CPCC score as a loss is well motivated. However, the paper does not fully demonstrate the effectiveness of embedding label hierarchy into the representation to downstream task. Many of the metrics are about whether the model learned label hierarchy or not. It is good that one could train a model as expected, but it is unclear what it is going to be useful for. The paper demonstrates the effectiveness on one-shot generalization for mid or coarser level classification tasks, but the task is somewhat superficial and may be easily solved by combining scores of finer-grained classifiers. Moreover, there are previous methods that regularizes deep representations differently, which should have been compared.

---

> ### Author Response · Authors · 2022-11-17
> **Response to Reviewer NCZE**
>
> *We want to thank the reviewer for pointing out related works and proposing better experiments. We have **included the suggested baselines**, and **redesigned the generalization experiments**. Additionally, we have found that **CPCC is useful for OOD detection** (Sec. 4.4 + Fig 3).*
>
> > While I do not closely follow this topic of learning with label hierarchy, it seems there are multiple previous works on deep representation learning with label hierarchy. For example, [Zhang et al., 2016], [Bertinetto et al., 2020].
>
> *Thanks for mentioning these papers! We agree they are important baselines and have included them in our experiments. CPCC is further added to each of them and shows extra improvements across all metrics. Moreover,, Bertinetto et al., 2020 confirm that hierarchical regularizers are rare. The only method is [1] (overlapping group lasso) which is tricky to adapt to modern networks. We hope this adds to the novelty of our method.*
>
> > Experiments on one-shot generalization is not particularly interesting nor significant. ... In this regard, more meaningful experiments would be one-shot or few-shot generalization to unseen classes, i.e., can we learn a fine-grained representation of class "0" by leveraging knowledge of its superclass ("even") and neighbors ("2, 4, 6, 8")?
>
> *Thank you for your suggestion on improving our generalization task design!*
>
> *We admit that “zero shot” is more reasonable when we have the prior knowledge that the train and test set have the same hierarchy. Therefore, in Table 1, we replaced Mid/Coarser Accuracy with “zero shot” performance per your suggestion. Adding CPCC consistently provides a performance boost, on all levels and objectives.*
>
> *We have additionally created splits of MNIST, CIFAR and BREEDS at the coarse level, into source and target, so that we can study generalization to unseen mid/fine classes. Please see the Data section (4.1) for details. New results are in Table 2.*
>
> *In the original draft’s Appendix C.2, we had reported experiments on BREEDS, for generalization to unseen classes in the target set. We were uncomfortable with the low overall performance (even though CPCC provided gains) so we didn’t put it in the main section. We have now searched over hyperparameters to get the best flat cross entropy performance, and used the tuned parameters for all objectives with and without CPCC for better comparison. Additional results on CIFAR/MNIST’s unseen task generalization were also added to Table 2.*
>
> *In Table 2, coarse/coarser levels are shared between source and target. Although there is a subpopulation shift, zero shot transfer can still be applied and CPCC provides a clear advantage. We then use one-shot learning to transfer knowledge to unseen mid/fine levels. We observe that using CPCC is helpful if the unseen class label is close to its coarse label (as is the case for LIVING17 and NONLIVING26), but that performance is worse if the new class is very fine-grained (all other datasets). We provide more explanation in Sec. 4.4.*
>
> *In terms of OOD applications, following a suggestion by reviewer SKmB, we have added OOD detection as an area where CPCC is very helpful (see Sec. 4.4 and Figure 3).*
>
> *As another side note, zero shot generalization is not applicable on generic label graphs. CPCC can still be applied there, and we leave this direction for future work.*
>
> [1] Zhao et al., 2011, Large-scale category structure aware image categorization.

---

> > ### Comment · Reviewer_NCZE · 2022-11-22
> > **Thanks for response**
> >
> > We thank authors for their response. Most concerns are resolved thanks to additional experiments. I am happy to raise my initial rating to 6.

---

### Official Review · Reviewer_EJjb · 2022-10-24

**Confidence:** 4
**Correctness:** 3
**Technical Novelty And Significance:** 2
**Empirical Novelty And Significance:** 2
**Recommendation:** 5

**Clarity, Quality, Novelty And Reproducibility:**

Clarity: overall good.

Quality: fair.

Novelty: weak.

Reproducibility: could be reproduced.

**Strength And Weaknesses:**

Strengths:

[+] The idea of preserving the class hierarchy in learning representations is interesting and sounds reasonable.

[+] The preliminaries give good background knowledge about the class hierarchy and the CPCC.

[+] The figure demonstrates clearly the class hierarchy.

Weaknesses:

[-] The motivation for embedding class hierarchy is not convincing. For instance, why does the model need to know whether a digit is even or odd? The pre-defined class hierarchy in MNIST is artificial, which would not make sense to the model. Intuitively, 6(even) is more similar to 9(odd) rather than 4(even).  Moreover, the class hierarchy would not always available in practice. The title of this paper lets readers expect to see how to explore the class hierarchy rather than just using it.

[-] The method is a bit short of important details about using the weights of edges in the tree graph to compute the weighted length of the shortest path.  Moreover, if there are several shortest paths between pairwise nodes in the general graph, how does the proposed tree metric deal with it?

[-] The paper argues that the tree metric is invariant to rotations of the tree and the changing of the root node. In practice, if the class hierarchy is pre-defined, is it reasonable to change the root node? With the class hierarchy, when does the tree metric work better than the dendrogrammatic distance in a real-world system?  If you wish to show the benefits of the proposed metric, extensive ablation studies are needed to compare the proposed method with other baselines embedding the class hierarchy.  And it would be better to provide more deep insights about the proposed metric.

[-] Is there any theoretical guarantee to make sure that the proposed regularization converges? (Especially the main body of this paper is the regularization.) Is the mini-batch training strategy equivalent to the designed strategy in Eq. (2) where the regularization is constructed on the whole dataset?

[-] In the experiments, due to the use of extra information about the class hierarchy, the better performance of the model with CPCC is not that surprising. It would be better to give more deep insights through the investigation of the regularization.

a). It would be more convincing to investigate the effects of CPCC on different objectives.

b). According to the Fine ACC and Coarse ACC in MNIST, the baselines with CPCC does not always outperform the baselines without CPCC. Here we need some explanations about the failure cases.

c). Based on the Mid Acc and Coarser Acc, this paper provides a good way to show the one-shot generalization. However, it is necessary for baselines with CPCC to show the differences before and after the fine-tuning.

d). What is the effect of the number of layers in the hierarchy classes in the CPCC?

e). Are there any benefits of regularization in solving the out-of-distribution problem?


**Summary Of The Paper:**

This paper develops a cophenetic correlation coefficient (CPCC) regularizer based on the given class hierarchy for classification.  The proposed regularizer is computationally lightweight and easy to implement. They demonstrated that the regularizer can learn more interpretable representations due to the preservation of the tree metric. Experiments six real-world datasets show that the proposed method achieves better in-distribution generalization as well as under sub-population shifts.

**Summary Of The Review:**

The reviewer thinks the paper is marginally below the acceptance threshold. The main contribution in the method section is the proposed metric. However, the paper needs to provide more theoretical analysis and ablation studies to show the effectiveness of the proposed metric.

---

> ### Author Response · Authors · 2022-11-17
> **Response to Reviewer EJjb Part 1**
>
> *We thank the reviewer for insightful comments. We have **added a brief discussion on the convergence of our objective function under SGD** in Appendix B. We also add **more ablation studies + empirical evaluations** to the experiment section.*
>
> > The motivation for embedding class hierarchy is not convincing. For instance, why does the model need to know whether a digit is even or odd? The pre-defined class hierarchy in MNIST is artificial, which would not make sense to the model. Intuitively, 6(even) is more similar to 9(odd) rather than 4(even).
>
> *We use the MNIST odd/even hierarchies mainly as a toy example to prove our method behaves as expected (i.e., fine classes are grouped by coarse level labels in the representation). As you point out, the most interesting setting is datasets where class hierarchy is not artificial, but visually meaningful. In these settings, the extra information from the hierarchy is also proved to be useful for better generalization results.*
>
> > Moreover, the class hierarchy would not always be available in practice. The title of this paper lets readers expect to see how to explore the class hierarchy rather than just using it.
>
> *We had not thought of that interpretation. If you think it necessary, we could add a word like Existing or Prior to the title, giving: Learning Structured Representations by Embedding Existing/Prior Class Hierarchies.*
>
> > The method is a bit short of important details about using the weights of edges in the tree graph to compute the weighted length of the shortest path. Moreover, if there are several shortest paths between pairwise nodes in the general graph, how does the proposed tree metric deal with it?
>
> *Any conventional shortest distance algorithm can be applied to the label hierarchy without issue. Having multiple valid shortest paths does not matter as the CPCC score only uses the value of the shortest path, which is unique.*
>
> > The paper argues that the tree metric is invariant to rotations of the tree and the changing of the root node. In practice, if the class hierarchy is pre-defined, is it reasonable to change the root node?
>
> *Yes, it can be helpful to properly handle changes to the root node. In practice data hierarchies are evolving. Different versions of the same dataset, which has a given hierarchy, introduce new coarse to fine relationships, for example, iNaturalist 2017-2019 -> 2021, Mapillary V1.2 -> V2.0, (see [4] for detailed discussion of this setting) to existing hierarchies. If there are new classes where the granularity of the new labels is unknown, or simply if they do not have a parent node in the previous version, it may be needed to rotate part of the tree or to change the root node depending on the granularity of the new labels. In this case, the representation structure of an existing model, trained with CPCC, could readily be fine-tuned to accommodate for the new labels, without needing to completely overhaul the embedding space.*
>
> > With the class hierarchy, when does the tree metric work better than the dendrogrammatic distance in a real-world system?
>
> *Besides being invariant to the root node which can be useful in a real-world system with shifting hierarchies, the tree metric $d_\mathcal{T}(.,.)$ can be applied to general label graphs (MetaShift [5]) where the least common ancestor is not defined.*
>
> > Is there any theoretical guarantee to make sure that the proposed regularization converges? (Especially the main body of this paper is the regularization.) Is the mini-batch training strategy equivalent to the designed strategy in Eq. (2) where the regularization is constructed on the whole dataset?
>
> *Note that in general, due to the non-convexity of the loss function with deep neural networks, one cannot hope to obtain convergence to the global optimal (if exists). Absent strict assumptions on the saddle points of Eq. (2), e.g., the strict saddle point property [1] that can be hard to verify in practice, in general, it is still unclear that gradient descent will always converge to a local minimum [2]. So instead, a more realistic convergence criterion that we can hope for is to show that when optimizing the objective function in Eq. (2), SGD will converge to a stationary point. To this end, under certain regularity conditions on the data, we show that SGD will converge to a stationary point of Eq. (2) with probability 1 in Appendix B, by invoking a key theorem in recent advances of nonconvex optimization with SGD (Theorem 2, [3]).*
>
> *Regarding the property of the mini-batch estimation of CPCC, it inherits the properties of the sample correlation coefficient, in particular that of being consistent, so for sufficiently large batch sizes, the approximation should be justified.*

---

> ### Author Response · Authors · 2022-11-17
> **Response to Reviewer EJjb Part 2**
>
> > In the experiments, due to the use of extra information about the class hierarchy, the better performance of the model with CPCC is not that surprising.
>
> *Indeed! Therefore we also included multiple baselines (multitask/curriculum/sum loss/etc.) which leverage the extra hierarchical label information. In Table 1, note that Flat+CPCC is still better than other non Flat baselines without CPCC, which was not necessarily expected.*
>
> > It would be more convincing to investigate the effects of CPCC on different objectives.
>
> *We want to emphasize that all baselines are different objectives, and for all of them we added CPCC as a regularization and observed performance gains. To also have a “non cross-entropy flavor” loss, we have added the quadruplet (generalized triplet) loss to our set of objectives. The result is consistent with others.*
>
> > According to the Fine ACC and Coarse ACC in MNIST, the baselines with CPCC does not always outperform the baselines without CPCC. Here we need some explanations about the failure cases.
>
> *The difference is negligible given standard deviation and the >99% performance . Also, as you have mentioned earlier, MNIST’s label hierarchy is artificial, so using CPCC may confuse the model by creating artificial groupings that do not leverage features helpful for classification.*
>
> > Based on the Mid Acc and Coarser Acc, this paper provides a good way to show the one-shot generalization. However, it is necessary for baselines with CPCC to show the differences before and after the fine-tuning.
>
> *We have made a significant revision to the generalization section 4.4 based on your comments. We now only fine tune when necessary. If the trained model can provide a class prediction without fine tuning (via appropriate marginalization of sub-classes, see e.g. Table 1) we don’t show the one-shot results anymore. Only the first 2 columns of Table 2, that concern generalization to unseen classes, require fine tuning (e.g. adding a giraffe fine class to the existing animal coarse class). In that case, we cannot simply use the pretrained classifier.*
>
> > What is the effect of the number of layers in the hierarchy classes in the CPCC?
>
> *That is an interesting question. In Appendix Fig. 4, we have added the distance matrix obtained when using a three layer CPCC (using fine/mid/coarse and fine/coarse/coarser levels). CPCC correctly embedded the extra level of information. The fine/coarse level performance is still similar and outperforms a model without CPCC.*
>
> > Are there any benefits of regularization in solving the out-of-distribution problem?
>
> *We did make a preliminary attempt at the OOD problem in the previous draft’s Appendix C.2, where we tried to see if structured representations can learn unseen fine classes on BREEDS.*
>
> *We have developed that direction more in the updated manuscript, Section 4.4 and Table 2 in particular. Table 2 shows that CPCC helps on mid levels and above, but hurts one-shot generalization to new fine classes on all but two datasets. We posit that CPCC tightly groups together the new fine classes with the ones from the same coarse class seen during training, and that this grouping makes it hard to linearly separate the new classes (this fact is supported by the gains observed on the CoarseAcc).*
>
> *Instead of solving the OOD problem, thanks to reviewer SKmB’s suggestion, we have realized that CPCC can be used for OOD detection. We show a significant advantage empirically at the end of Sec. 4.4 and in Figure 3.*
>
> [1] Lee et al., 2016, Gradient Descent Converges to Minimizers.\
> [2] Murty and Kabadi, 1987, Some NP-Complete Problems in Quadratic and Nonlinear Programming.\
> [3] Patel and Zhang, 2021, Stochastic Gradient Descent on Nonconvex Functions with General Noise Models.\
> [4] Lin et al., 2022, LECO: Learning with an Evolving Class Ontology.\
> [5] Liang and Zou, 2022, Metashift: A Dataset of Datasets for Evaluating Contextual Distribution Shifts and Training Conflicts.

---

> ### Author Response · Authors · 2022-12-01
> **Thanks for your feedback**
>
> We thank you once again for the feedback. We encourage you to reassess our work in light of our responses and updates, and reach out with any remaining concerns and questions; we are happy to continue the discussion.

---

### Official Review · Reviewer_SKmB · 2022-10-29

**Confidence:** 4
**Correctness:** 3
**Technical Novelty And Significance:** 2
**Empirical Novelty And Significance:** 2
**Recommendation:** 5

**Clarity, Quality, Novelty And Reproducibility:**

The paper is clear and well-written and looks reproducible. The proposed method is simple, and I think the novelty would not be an issue if the authors could provide more empirical evaluation and fair comparisons.

**Strength And Weaknesses:**

Strengths:

- The paper follows an interesting direction, which has good potential.
- The proposed method is simple and makes sense.
- The paper is well-written and easy to follow.

Weaknesses:

- The authors aim to learn structured representations. However, there is no quantitate evaluation to show that their method results in more structure in the representation space. They only show a t-SNE plot of their method and compare it with the baseline, which is a good sign but not a thorough evaluation. They could evaluate, for instance, the OOD task, which benefits from the more structured representation [1].

- The evaluation datasets are limited. It would be good to show some results on ImageNet(or at least a smaller version of it, such as Tiny Imagenet.)

- The paper misses some baselines. It would be better to compare with $t(.,.)$ in table 1. Moreover, comparing with [1] would be interesting.

-  I am a bit confused by the term ERM. If I am not mistaken, ERM stands for empirical risk minimization, and the authors refer to Eq 3. by ERM, which is the cross-entropy loss. But all losses in this paper minimize empirical risk. Also, why it's performance is too low in Table 1.?

[1] Hoffmann et al. Ranking Info Noise Contrastive Estimation: Boosting Contrastive Learning via Ranked Positives.





**Summary Of The Paper:**

This paper introduces a new method for learning structured representations from hierarchical annotations. A typical approach for learning representations in a supervised setup is to train a classifier that assigns a single label to a given image. This approach, however, might result in suboptimal representations as samples from all two classes are treated similarly, i.e. misclassifying a specific class of dog with another dog is penalized similarly to misclassifying it as a cat. The proposed method uses freely available hierarchical semantic annotation to enhance the learned representations. Their proposed method involves a new distance metric of two nodes in the hierarchical semantic tree used as an auxiliary loss function that preserves similarity between a pair of given samples on top of standard cross-entropy for classifying them.

**Summary Of The Review:**

Overall, the proposed method is simple and makes sense. But it needs to be supported by a better and fairer experimental setup.

---

> ### Author Response · Authors · 2022-11-17
> **Response to Reviewer SKmB Part 1**
>
> *We thank the reviewer for suggesting additional experimental comparison. For extra empirical results, we have added the result of the **OOD task** you suggested (See end of Sec. 4.4 / Figure 3), and **compared CPCC with other newly added baseline methods** (See HXE/Soft/Quad in the baseline section 4.2). We do not include RINCE’s results in the paper for reasons explained later.*
>
> > There is no quantitative evaluation to show that their method results in more structure in the representation space. They only show a t-SNE plot of their method and compare it with the baseline, which is a good sign but not a thorough evaluation. They could evaluate, for instance, the OOD task, which benefits from the more structured representation.
>
> *CPCC and silhouette scores, especially silhouette evaluated on coarse level, are quantitative hierarchical metrics that were reported in Table 1. The OOD task is an indirect downstream method for evaluation, however, we do like the idea, thank you for the great suggestion. We have added it at the end of Sec 4.4. (results in Figure 3). CPCC shows a strong advantage for the detection of outliers (gains around 10% across all methods).*
>
> > The evaluation datasets are limited. It would be good to show some results on ImageNet (or at least a smaller version of it, such as Tiny Imagenet.)
>
> *BREEDS is a subset of ImageNet which contains a proven visually meaningful hierarchy in our experiments. We avoid directly using ImageNet (its hierarchy might be inferred from Wordnet) because Wordnet graphs contain non semantically meaningful relationships. This last point is discussed in the BREED’s paper.*
>
> > The paper misses some baselines. It would be better to compare with $t(.,.)$ in table 1. Moreover, comparing with RINCE would be interesting.
>
> *In our experiments, due to the tree structure of the hierarchy, it is equivalent to use $t(.,.)$ and $d_\mathcal{T}(.,.)$, as we explained in Prop. 3.2 and the paragraph after the proposition.*
>
> *For the comparison with RINCE, it’s hard to directly compare it with our method because contrastive learning based methods, even supervised, learn the final classifier after learning the representation, while we learn the classifier and the representation together. Second, contrastive learning methods can be influenced by augmentation methods, which makes it a bit hard to compare accuracy results on equal footsteps.*
>
> *Out of curiosity, as an attempt for fair comparison, we trained RINCE with ResNet18 ([1] used ResNet50) as we did for all methods in our paper. We used their default training parameters except for changing the encoder. The results are evaluated on CIFAR100. See Table below.*
>
> | Experiments           |  CPCC | silhouette |  OOD  | FineAcc | CoarseAcc |
> |-----------------------|:-----:|:----------:|:-----:|:-------:|:---------:|
> | RINCE-in              | 12.15 |    -4.95   | 56.05 |  74.42  |    85.4   |
> | RINCECPCC-in          | 12.74 |    -4.29   | 54.51 |   73.40  |    85.2   |
> | RINCECPCC-unit-in     | 11.22 |    -4.08   | 57.09 |  72.89  |    85.1   |
> | RINCE-out             | 10.56 |    -5.13   | 55.36 |  74.09  |    84.8   |
> | RINCECPCC-out         | 11.22 |    -4.11   | 56.63 |  73.19  |    85.3   |
> | RINCECPCC-unit-out    | 11.81 |    -4.57   |  55.80 |  73.47  |     85.0    |
> | RINCE-in-out          | 11.04 |    -5.58   | 57.89 |  74.59  |    84.9   |
> | RINCECPCC-in-out      | 11.75 |    -3.80    | 55.19 |  73.51  |    85.3   |
> | RINCECPCC-unit-in-out | 13.07 |    -4.19   | 55.54 |  72.96  |    84.8   |
>
> *First, the accuracy is expectedly lower than the result in their table on OOD/Fine Accuracy and Coarse Accuracy, possibly due to the change of encoder. Second, CPCC and silhouette scores are also very low even compared to the naive cross entropy.
> We tried to add CPCC on (i) the unit hypersphere representation (ii)  the backbone encoder’s representation, but it had limited change on the embedding structure.*
>
> *Therefore, we suspect the learned RINCE representation still lies on the side of uniformity (in terms of representation structure) which is helpful for fine level classification but not really on hierarchical metrics. Also, interestingly, note that the differences between the in, out and in-out training objectives is reflected in the CPCC score, because “in” tends to capture highly similar pairs (fine class within coarse group) which aligns with our motivation for CPCC.*

---

> ### Author Response · Authors · 2022-11-17
> **Response to Reviewer SKmB Part 2**
>
> > I am a bit confused by the term ERM. If I am not mistaken, ERM stands for empirical risk minimization, and the authors refer to Eq 3. by ERM, which is the cross-entropy loss. But all losses in this paper minimize empirical risk. Also, why it's performance is too low in Table 1.?
>
> *Sorry for the confusion, indeed ERM might not have been the best choice. We have replaced it with Flat everywhere. We conjecture that the Flat cross entropy loss has a slightly lower performance than other methods, because they all leverage in some way hierarchical information about the class structure (i.e. extra label information): coarse labels can be helpful for fine level classification tasks.*
>
> [1] Hoffmann et al., 2022, Ranking Info Noise Contrastive Estimation: Boosting Contrastive Learning via Ranked Positives.

---

> ### Author Response · Authors · 2022-12-01
> **Thanks for your feedback**
>
> We thank you once again for the feedback. We encourage you to reach out with any remaining concerns and questions, and we are happy to continue the discussion.

---

### Author Response · Authors · 2022-11-17
**Summary of Revisions**

We thank all reviewers for their suggestions and their appreciation of our work. We have made several major revisions (now highlighted in blue) including:
- **Adding three baselines**: Hierarchical Cross Entropy [1], Soft Label [1] and Quadruplet Loss [2]. The conclusions are similar to the ones with previous baselines. Adding CPCC results in improvements across all metrics.
- **Reframing the generalization task**. We removed one-shot mid/coarser results in Table 1 by replacing them with zero shot results using a simple marginalization. We have split datasets in the experiments reported in Table 2, to better study “out-of-hierarchy” generalization. Note that in that case, one-shot learning is necessary to learn unseen mid and fine classes. We find that CPCC is mostly useful for mid classes and above but has limitations on fine level generalization. Also, note that in the in-distribution setting, by reusing the output of fine grained classifiers, we obtain much better performance on mid/coarser level compared to one-shot transfer (reported in the initial submission).
- **Adding OOD detection** [3] results which demonstrate another advantage conferred by CPCC (end of Section 4.4 and Figure 3).
- Moving some MNIST and BREEDS results to Table 4 in the appendix to accommodate the page limit. We only include the cross entropy loss (formerly called ERM, and now renamed Flat for clarity) with or without CPCC for MNIST and BREEDS in the main paper.

We are running the new objectives on BREEDS and MNIST and the results will be added to the appendix if time permits. However, we do not expect significant differences from Flat vs. FlatCPCC, as shown in the original submission and on CIFAR100, so our conclusions should remain the same.

[1] Bertinetto et al., 2020, Making Better Mistakes: Leveraging Class Hierarchies With Deep Networks. \
[2] Zhang et al., 2016, Embedding Label Structures for Fine-Grained Feature Representation. \
[3] Hoffmann et al., 2022, Ranking Info Noise Contrastive Estimation: Boosting Contrastive Learning via Ranked Positives.

---

### Decision · Program_Chairs · 2023-01-20

**Decision:**

Accept: poster

**Justification For Why Not Higher Score:**

The experiments were initially insufficient, though the rebuttal strengthened them. The novelty was fine.

**Justification For Why Not Lower Score:**

The overall quality is good.

**Metareview: Summary, Strengths And Weaknesses:**

Four experts reviewed the paper. Two reviewers were negative about the paper, with concerns about novelty and experimental analyses. Reviewer NCZE was initially concerned about the experiments, but the authors' rebuttal addressed it well. The last reviewer was excited about the paper and rated it "good paper". AC decided to recommend acceptance because the experiments, which the first two reviewers found weak, were strengthed by the rebuttal, and the novelty concern was not concrete. The authors are encouraged to make the necessary changes to the paper to the best of their ability following the reviewers' comments.

**Note From Pc:**

if the above contains the word "oral" or "spotlight" please see: "oral" presentation means -> notable-top-5% and "spotlight" means -> notable-top-25%. As stated in our emails, we are disassociating presentation type from AC recommendations